mechanical engineering

image decomposition, full-field deformation, model validation, computational modelling, condition monitoring

**Author for correspondence:**
W. J. R. Christian
e-mail: w.j.r.christian@liverpool.ac.uk

# Comparing full-field data from structural components with complicated geometries

W. J. R. Christian[1], A. D. Dean[2], K. Dvurecenska[1], C. A. Middleton[1] and E. A. Patterson[1]

[1]School of Engineering, and [2]Institute of Infection, Ecological and Veterinary Sciences, University of Liverpool, Liverpool, UK

 WJRC, 0000-0003-3638-7297; ADD, 0000-0001-9033-7560; KD, 0000-0003-0642-1095; CAM, 0000-0001-9488-9717; EAP, 0000-0003-4397-2160

A new decomposition algorithm based on QR factorization is introduced for processing and comparing irregularly shaped stress and deformation datasets found in structural analysis. The algorithm improves the comparison of two-dimensional data fields from the surface of components where data is missing from the field of view due to obstructed measurement systems or component geometry that results in areas where no data is present. The technique enables the comparison of these irregularly shaped datasets without the need for interpolation or warping of the data necessary in some other decomposition techniques, for example, Chebyshev or Zernike decomposition. This ensures comparisons are only made between the available data in each dataset and thus similarity metrics are not biased by missing data. The decomposition and comparison technique has been applied during an impact experiment, a modal analysis, and a fatigue study, with the stress and displacement data obtained from finite-element analysis, digital image correlation and thermoelastic stress analysis. The results demonstrate that the technique can be used to process data from a range of sources and suggests the technique has the potential for use in a wide variety of applications.

## 1. Introduction

Optical measurement techniques can provide high spatial resolution data describing displacements, deformation, strain and stress which are captured on the surface of structures with minimal surface preparation and without the bonding of sensors. This has made optical techniques attractive for studying the behaviour of safety critical structures, such as those found in the aerospace sector, where sensors such as strain gauges or accelerometers may change structural dynamics or limit the

quantity of information that can be obtained from a component. Optical techniques are often used as part of the process of validation of computational models and for exploring damage mechanics [1]. These highly instrumented tests yield substantial quantities of data that must be carefully processed to ensure that subtle features in the data are not missed. This can be both analytically and computationally challenging and thus, it is common for full-field data to be used in a qualitative manner, where the analyst makes a subjective judgement about agreement, or values are extracted from only a handful of locations and pointwise comparisons are made. These pointwise comparisons between datasets are common in industry, stemming from practices established with accelerometers and strain gauges where the agreement is determined using simple statistical tests [2].

When performing modal analysis, techniques exist for determining the agreement between modal data from different sources, with perhaps the most commonly encountered technique being the modal assurance criterion [3]. The strength of this criterion is that it can be applied to experimental data from a wide range of sources, including full-field optical measurement techniques [4], but it can only be used to compare dynamic behaviour. The result of the modal assurance criterion is a value between 0 and 1 indicating the quality of agreement. This normalization makes it difficult to define an acceptance criterion as the criterion output cannot be related to the measurement uncertainty of the experiment [5]. Others have explored using singular value decomposition to compare mode shapes. These approaches reduce the dimensionality of the data by projecting it onto an orthogonal set of kernels resulting in a feature vector with the same units as the originally measured or simulated data-field [5]. This feature vector is therefore more readily related to the measurement uncertainty of the experiment.

Another approach is image decomposition, where the dimensionality of the data is reduced by projecting it onto a set of orthogonal polynomials. These polynomials are not specific to a particular dataset and can be used for straightforward comparisons of data from experiments and models in quasi-static [6] and dynamic scenarios [7]. This makes the technique more generally applicable than modal assurance criterion and singular value decomposition techniques. Once decomposed, a dataset that may have originally contained $10^4$ to $10^6$ data points, is instead represented using a feature vector that consists of typically tens or hundreds of terms. This simplifies the process of comparing one dataset with another when performing validation or damage assessments. This approach to processing and comparing applied mechanics data has been written into a European Committee for Standardisation (CEN) workshop agreement [8].

A significant drawback of standard image decomposition techniques is encountered when decomposing data that is irregularly shaped, such as when an entire component is assessed as opposed to a small region of interest [9]. This is because the polynomials used for the decomposition are only orthogonal over a particular domain and this orthogonality is lost when the shape of the data deviates from that domain [7]. For example, Zernike polynomials require a circular domain [10], while discrete Chebyshev polynomials require a rectangular domain [6]. If data is missing from within the domain then the polynomials are not orthogonal when the data is projected onto them and thus it is not possible to reconstruct the data from the feature vector [11]. One approach to this problem is to warp or deform the data so that its shape matches the domain of the polynomials using techniques such as conformal mapping [9]. This works when the data-field does not contain holes and has been used to decompose rectangular data fields using Zernike polynomials [12]. As the dataset is warped, the spatial resolution of the data will vary in different locations and thus the feature vectors may be more sensitive to shapes and patterns present in areas with few data points than in areas with a high density of datapoints. Another approach is to interpolate the data so that the resulting datapoints match the domain of the polynomials [9]. Often these interpolated datapoints are approximately equal to the mean of the real datapoints surrounding the area and thus could increase the perceived similarity between the data-fields when comparisons are made.

A mathematically rigorous approach to irregularly shaped datasets is to modify the polynomials so that they are orthogonal on the domain of the dataset. Wang *et al*. [7] used this approach to generate adaptive geometric moment descriptors (AGMDs) which used Gram-Schmidt orthogonalization to convert monomials into polynomials that were orthogonal for a particular shape. This approach means the data does not need to be warped to correspond to the shape of the polynomial domain and limits the distances over which data must be interpolated. However, it does not eliminate the need for interpolation as both data-fields still need to have datapoints at the same locations in order for them to be decomposed using the same polynomials. In addition to this, it can be a labour-intensive task for the analyst, as the process has to be tailored for different irregular shapes. This makes it difficult to automate the comparison process and thus limits its application in industry. The Gram-Schmidt orthogonalization algorithm is also numerically unstable due to rounding errors resulting in the algorithmically computed

basis vectors being in practice only nearly orthogonal [13]. Salloum *et al*. [14] have also explored the use of Gram-Schmidt orthogonalization to obtain an orthogonal set of polynomials for decomposing data, demonstrating it for both two-dimensional and three-dimensional datasets. However, their approach results in the two datasets being represented by similar but non-identical sets of polynomials and thus are not strictly comparable. Furthermore, their final assessment is made by a subjective comparison between two lines plotted on a graph.

This paper introduces a new algorithm based on QR factorization that allows comparisons between irregularly shaped data-fields while eliminating the need for interpolation or warping of data fields. This simplifies the comparison process allowing greater automation. To explore the behaviour of this algorithm, it is applied to three distinct case studies and the results are discussed.

# 2. Numerical method

## 2.1. Pre-processing data

Before two datasets can be decomposed, it is necessary to align them such that they have a common coordinate system. This is because experimental techniques generate data with real world coordinates, where the origin is normally arbitrarily defined during equipment set-up and calibration, whereas simulations typically generate data where the coordinate system is defined based on the component or sub-assembly being modelled. In ideal circumstances, some kind of fiduciary mark will be visible on the component surface so that the coordinate systems can be perfectly aligned, but this is not always feasible and thus manual alignment is typically necessary [15]. In this study, this was achieved through manual translation and rotation of one dataset until the two were in alignment, a qualitative check was then performed to ensure that the data was well aligned. This is not an ideal method of alignment but was unavoidable as this study uses data from prior studies where the simulation and experiments had different coordinate systems.

Once the datasets were aligned, they were projected onto a two-dimensional plane to remove the third spatial dimension. If the datasets have significant perturbations in the third spatial dimension, then multiple datapoints could potentially be projected to the same location on a two-dimensional plane. In this situation, conformal mapping procedures can be used to flatten the data without significantly warping it [7]. This was not necessary for any of the examples used in §3.

After alignment and flattening, further pre-processing of the data was required to ensure the two datasets cover the same domain. It was assumed that each dataset contained datapoints that were approximately evenly distributed across space and thus the spatial resolution could be calculated as a function of the area covered by that dataset and the number of datapoints it contains. To calculate the area, a bounding polygon shape was first found for each dataset using the alpha shape algorithm [16]. Other algorithms could be used for this purpose without significant effects on the results. The alpha shape algorithm was used as it is a common algorithm built into MATLAB and other programming languages that can determine the outer boundary of a shape, as well as inner boundaries that may be caused by large internal regions lacking data. It is also computationally efficient, taking less than 400 ms on a conventional laptop computer to process the largest dataset used in this study. Once the bounding shape was determined, the spatial resolution $h_i$ of a dataset $i$ was estimated as

$$h_i \approx \frac{\sqrt{A_i}}{\sqrt{N_i - 1}},$$

(2.1)

where $A_i$ is the area of the bounding shape of dataset $i$ and $N_i$ is the number of datapoints in dataset $i$. Any datapoint in the first dataset that did not have at least one member of the second dataset within a distance $h_1$ was then discarded. Similarly, any datapoint in the second dataset that did not have at least one member of the first dataset within a distance $h_2$ was also discarded. This results in two datasets that can then be compared using decomposition methods.

## 2.2. Decomposition using QR factorization

In overview, the decomposition method works by first computing an orthogonal basis for each dataset such that the data can be decomposed into feature vectors. These basis sets were computed from monomials that had been orthogonalized using QR factorization. In order to make the feature vectors comparable, one of them was then subject to a basis transformation such that it had the same basis as

the other feature vector. These feature vectors could then be used for validation or condition monitoring processes. The detailed method is described as follows. An applied mechanics dataset can be considered as three column vectors of length $N$, where $N$ is the number of datapoints in the dataset. These columns are the data $\mathbf{d}$ and the corresponding two-dimensional coordinates $\mathbf{x}$ and $\mathbf{y}$. If $\mathbf{x}$ and $\mathbf{y}$ represent a rectangular grid, then a set of $K$ basis vectors can be calculated using, for example, discrete Chebyshev polynomials [17] or cosine functions [18]. These basis vectors are also of length $N$, and we denote them by $\mathbf{p}_j$, with $j = 0, \ldots, K-1$ and $K \leq N$. Thus the dataset can be written as

$$\mathbf{d} = P\mathbf{a}, \tag{2.2}$$

where $P$ is a $N \times K$ orthogonal matrix in which the $j$th column is $\mathbf{p}_j$, and $\mathbf{a}$ is a feature vector of length $K$. This enables the dimensionality of the dataset to be reduced by truncating the feature vector $\mathbf{a}$ at a particular length. After truncation, $\mathbf{a}$ can be padded with zeros so that it has the length $K$ and can thus be reused in equation (2.2) to obtain an accurate representation of the original column vector of data, $\mathbf{d}$.

However, on irregular domains the basis vectors $\mathbf{p}_j$ are not orthogonal; this can be rectified using QR factorization via the MATLAB function qr($P$,0). This routine writes $P = QR$, such that $Q$ is an $N \times K$ orthogonal matrix (i.e. satisfies the relationship $Q^T Q = I$), and $R$ is a $K \times K$ change-of-basis matrix. Using the matrix $Q$, the data are thus represented by

$$\mathbf{d} = Q\mathbf{b}, \tag{2.3}$$

with $\mathbf{b} = R\mathbf{a}$. The columns of $Q$ therefore form an orthogonal basis for the dataset. In particular, the transformed feature vector $\mathbf{b}$ is now readily calculated as

$$\mathbf{b} = Q^T \mathbf{d}. \tag{2.4}$$

Note that the MATLAB function qr($P$,0) does not produce the full QR factorization, but the extra columns of $Q$ and rows of $R$ calculated by the latter method are not required for the present purpose and so can be disregarded, saving computational effort; if $N = K$ then the two methods are the same.

In order to compare two datasets $\mathbf{d}_1$ and $\mathbf{d}_2$, we therefore carried out the above process on each dataset in turn. This yielded the two orthogonal bases $Q_1$ and $Q_2$, with associated feature vectors $\mathbf{b}_1$ and $\mathbf{b}_2$. Crucially, the two bases are distinct from one another, meaning that one feature vector must be transformed into the basis of the other before they can be compared. This is easily done, as the QR factorization also provides the change-of-basis matrix $R$. Choosing to transform the second feature vector, we obtained

$$\mathbf{b}_2' = R_1 R_2^{-1} \mathbf{b}_2, \tag{2.5}$$

where we first multiplied $\mathbf{b}_2$ by $R_2^{-1}$ to transform back to the initial basis $P$, and then by $R_1$ to transform to the basis $Q_1$. Hence $\mathbf{b}_1$ could be meaningfully compared to $\mathbf{b}_2'$. Note that the two feature vectors must be the same length for this process to work, i.e. the value of $K$ is common to both factorizations. Finally, both feature vectors are scaled so that the differences between them are relatable to the measurement uncertainty of the experimental data,

$$f_1 = \frac{\mathbf{b}_1}{\sqrt{N_1}}, f_2 = \frac{\mathbf{b}_2'}{\sqrt{N_1}}, \tag{2.6}$$

where $N_1$ is the number of datapoints in the first dataset.

When performing validation the intention is to confirm that the simulation data is an accurate representation of reality as observed during an experiment. Thus, it is recommended that the feature vector representing the simulation should be the second dataset and thus be the one to undergo the basis transformation in equation (2.5). Similarly, for condition monitoring the purpose is to confirm that the behaviour of the structure is still the same as at the start of the test and therefore the experimental dataset representing initial condition should be the first dataset in the algorithm and thus the least processed.

## 2.3. Choosing the initial basis

In theory, any suitable set of linearly independent vectors may be chosen for the initial basis $P$, provided they are consistently defined on the spatial domains of both datasets. In practice, this is most readily achieved by choosing polynomial functions $p_j(x,y)$, $j = 0, \ldots, K-1$, and evaluating them on both

$x_1, y_1$ and $x_2, y_2$. Throughout the present work, we chose monomial functions $x^k y^l$, for integers $k$ and $l$ such that $0 \leq k + l \leq O_{\max}$, where $O_{\max}$ is the user-defined maximum order of the polynomial basis.

Care must also be taken that the polynomial basis does not exceed the effective resolution in each spatial direction. To calculate this, we found the smallest rectangular domain that contained every point in the dataset and superimposed a square mesh with approximately the same spatial resolution as the dataset. Defining $N_x$ and $N_y$ to be the number of points in the $x$ and $y$ directions, and recalling that the spatial resolution $h$ of a dataset is given approximately by (1), we have

$$
\left.
\begin{aligned}
N_x &\approx \mathrm{floor}\left[\frac{\max(x) - \min(x)}{h}\right] \\
N_y &\approx \mathrm{floor}\left[\frac{\max(y) - \min(y)}{h}\right]
\end{aligned}
\right\}.
\tag{2.7}
$$

Thus, we also imposed the constraints $0 \leq k \leq N_x$ and $0 \leq l \leq N_y$. If alternative polynomials are chosen for the initial basis, then the same constraints should be applied, with $k$ and $l$ simply representing the maximum powers of $x$ and $y$ in each polynomial basis function.

## 2.4. Quantifying reconstruction quality

The global representation error, a value quantifying the discrepancy between the reconstructed and original dataset, was obtained by calculating the root mean squared error between the original data and its reconstruction from the feature vectors. The reconstructed datapoints were at the same locations as the original datapoints and thus it was trivial to calculate the root mean squared error between the values. The aim when decomposing any dataset is for the representation error to be less than the measurement uncertainty of the system used to obtain the experimental data [8]. Ideally, this measurement uncertainty should be obtained from a rigorous calibration experiment where a specimen experiences known deformations while being measured using an identical measurement system [19]. This is so that all of the uncertainties related to an optical measurement system are incorporated into a single value.

While the global representation error of a dataset may be acceptable, at certain locations the original and reconstructed data may deviate substantially due to high gradients in the full-field data. For example, the stresses in a cracked specimen under tension rapidly increase with proximity to the crack tip while far away the stress field is nominally uniform. To quantify the local representation quality, most previous studies have focused on identifying clusters of datapoints where the absolute difference between the original and reconstructed datapoints are above a threshold, such as three times the global representation error [8]. For regularly defined domains where datapoints are on a grid, clusters can be found by identifying areas where adjacent datapoints are all above this threshold. This is difficult for irregularly defined data domains as adjacency has no clear definition. For this study, a new algorithm was developed to identify clusters of poorly represented datapoints for irregularly shaped datasets.

To efficiently identify which datapoints were adjacent to each other, a bounding shape was first found for the dataset. This bounding shape was calculated using the same alpha shape algorithm used in §2.1. The bounding shape for each dataset needed to be recalculated as datapoints on the boundary may have been removed while performing the overlap assessment described in §2.1. Once the bounding shape was determined, a Delaunay triangulation was calculated for the inside of the shape, such that each datapoint was associated with a vertex in the resulting triangulation. The Delaunay algorithm avoids fitting triangles to the datapoints that have high aspect ratios, so that each triangle could be used to determine adjacency. This resulted in a set of triangles, where each vertex represented a datapoint and each edge connected datapoints to nearby datapoints.

The representation error of each datapoint was then checked to see if it was greater than three times the measurement uncertainty. If all of the datapoints represented by the vertices of a triangle were above this threshold then the datapoints at the vertices were considered poorly reconstructed. All of the triangles consisting of three poorly reconstructed datapoints were checked to establish if any of them were connected by a common vertex. If they shared a common vertex then they were joined to make a polygonal shape where every vertex within it represented a poorly reconstructed datapoint. These polygonal shapes were equivalent to the clusters of poorly reconstructed datapoints that can be identified when the datapoints are on a regularly defined grid using the method described in [8]. The algorithm was applied to both datasets to identify large polygonal areas representing poorly reconstructed datapoints. If any cluster of datapoints was found to be larger than 0.3% of the total number of datapoints in the dataset then the dataset was considered to have a poor local reconstruction.

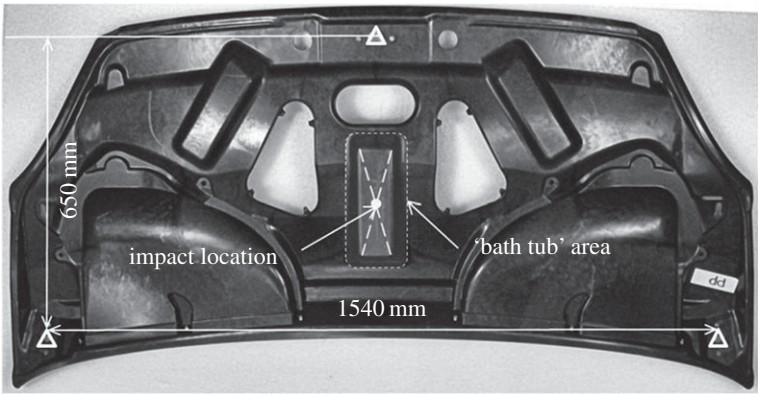

**Figure 1.** A photograph, taken from [20], showing the impacted surface of the bonnet liner and the impact location. The liner was mounted to a rigid frame at the locations marked by triangles.

This was the same threshold used for datasets with regularly defined domains described in a CEN workshop agreement on validation [8]. If the quality of the local reconstruction was insufficient for performing validation or condition-monitoring, then the size of the clusters of poorly reconstructed datapoints were reduced by increasing the maximum order of the monomials used during the decomposition process.

# 3. Case studies

Data from three different experiments were used to test the capabilities of the new decomposition algorithm. As this paper is focused on the processing of data, the following subsections briefly detail how the data was obtained and the limitations of the data processing in previous studies. For more details, the reader is directed to the sources cited.

## 3.1. Bonnet liner impact

A gas-gun was used to fire a projectile at a car bonnet liner [20]. The bonnet liner had been developed for a concept car and was manufactured from polyamide reinforced with chopped glass fibres. The liner had a complicated three-dimensional shape, with multiple cut-outs and holes and a non-rectangular boundary, as seen in figure 1. The liner was fixed to a rigid frame at a single location at its top and two locations at its bottom and a gas gun was aimed at the centre of the liner. A high-speed stereo digital image correlation (DIC) system (Q-450, Dantec Dynamics, Germany) was used to measure out-of-plane displacements on the surface of the liner during the impact event. In parallel to this experiment, the liner was simulated using ANSYS-LS-DYNA. The simulation represented the bonnet liner with 22 328 shell elements with the rigid supporting frame represented using simple boundary conditions. The impact event was simulated and out-of-plane displacements at the nodal locations were output. In the previous study, the data from both the simulation and experiment were then processed using AGMDs in order to perform a validation of the simulation.

In this study, a single timestep was used at 23.4 ms after the impact to illustrate the new decomposition algorithm. This timestep was chosen as it was at a point in time when the whole liner was in motion but before those motions had appreciably decayed. The experimental data were processed as the first dataset in the algorithm described in §2.2, and the simulation data was the second dataset. The rigid frame occluded the view of the DIC system of the top and bottom of the bonnet and thus experimental datapoints were not available at these locations. The three-dimensional shape of the bonnet liner also resulted in locations at which displacement data was unavailable. These locations can be identified from the plot at the top of figure 2. The measurement uncertainty of the DIC system must be known to ensure that the datasets are represented with enough polynomials to conduct a reliable validation process. To estimate this uncertainty, the same high-speed DIC system was used in a calibration experiment giving a relative uncertainty of 1.0% [19]. This relative uncertainty expresses the measurement uncertainty of the system as a percentage of the range of the measurand, thus the absolute measurement uncertainty for the dataset can be estimated by multiplying this relative uncertainty by the range of the measured

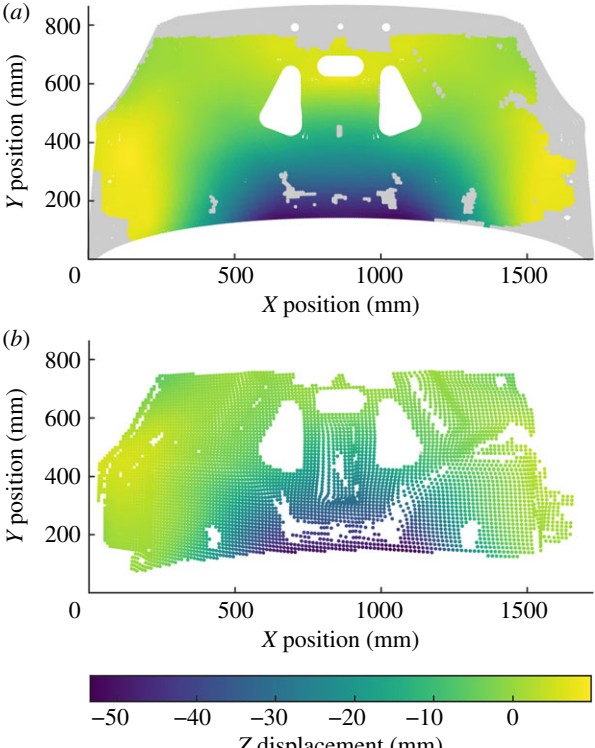

**Figure 2.** Aligned out-of-plane displacement fields from the simulation (*a*) and experiment (*b*), with simulation data removed during alignment shown in grey.

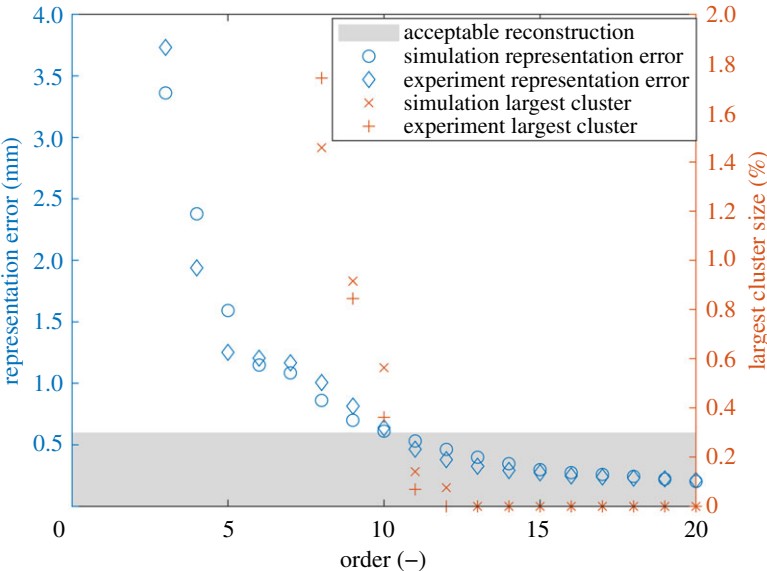

**Figure 3.** The effect of increasing order of polynomials on the representation error and largest cluster size for the bonnet liner simulation and experiment datasets.

out-of-plane displacements from the experiment. This results in an absolute measurement uncertainty of 0.60 mm. The representation error and size of the largest cluster of poorly represented datapoints are shown in figure 3 demonstrating a near monotonic decrease in both characteristics as the maximum order of the polynomial used increases. Polynomials up to a maximum order of 11 were used to achieve a representation of the data that fulfils the quality criteria described in §2.4.

After the data had been decomposed, two feature vectors, one representing the experimental data and the other representing the simulation data, were obtained. These feature vectors were then used as part of

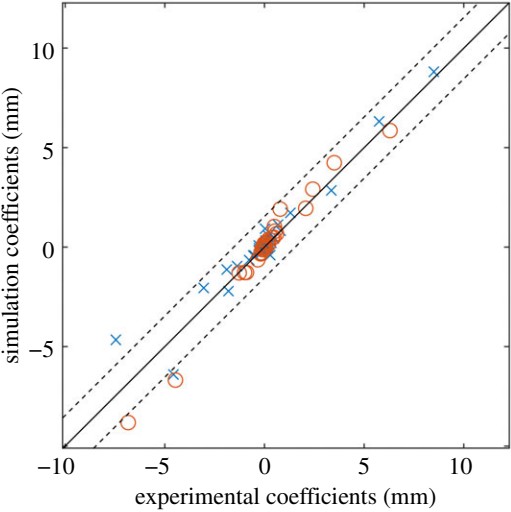

**Figure 4.** Validation diagram showing the agreement between the bonnet liner experiment and its simulation. The dashed lines indicate the acceptance region that should contain all of the points if the simulation is to be considered acceptable. The circles represent coefficients calculated using adaptive geometric moment descriptors, while the crosses represent coefficients calculated using the new algorithm.

a quantified validation methodology [8], whereby the feature vectors were plotted against each other as a scatter diagram, shown in figure 4. The dashed lines in the figure show the acceptance region defined by the expanded uncertainty, which was calculated as the vector sum of the measurement uncertainty of the DIC system and the global representation error of the experimental dataset, which is equal to the root mean squared error between the reconstructed and original data [8]. If all of the points on the diagram lie within the acceptance region then the model can be considered acceptable. In this case, two points are outside of this region and thus the simulation should not be accepted without further investigation. To compare the new decomposition algorithm with previous methods, the AGMDs calculated in [20] were plotted on the same axes as the coefficients from the new algorithm, shown in figure 4. The magnitudes of the coefficients are similar but the locations of the points corresponding to each algorithm are different as the two algorithms used different bases. When considering the acceptance region, two of the AGMD coefficients were outside of the acceptance region, which is the same finding as was obtained using the new algorithm.

## 3.2. Aerospace panel modal analysis

A prototype aerospace panel milled from a single block of 7075 aluminium was used for the second case study. The panel was flat on its front face but on the back had stiffeners and hole reinforcements milled into it, as shown in figure 5. The panel was simply supported by strings and a shaker attached near its bottom left corner [21]. The panel was excited at its first three harmonic frequencies and the out-of-plane displacement mode shapes measured using a stereoscopic DIC system (Q-400, Dantec Dynamics, Germany) using a pulsed laser to illuminate the panel and 'freeze' the motion. The panel was also simulated using a modal frequency response analysis performed in Altair Optistruct. In the simulation, the panel was modelled using 170 000 hex cell elements with the loads applied at the same location as in the experiment.

In the previous study, discrete Chebyshev polynomials were used and thus only a rectangular portion of the data was decomposed, resulting in experimental data from approximately 10% of the surface of the component being discarded. This rectangular area is shown using the dashed line in the top part of figure 5. For this study, all of the available DIC data was used, with only the simulation data around the edges of the panel discarded due to a lack of corresponding experimental data. The experimental and simulation data, after non-overlapping simulation data had been removed, is shown in figure 6. The relative measurement uncertainty of the DIC system was measured and found to be 1.75% [19]. Polynomials up to a maximum order of five were used to decompose the data such that an adequate global and local representation was achieved for the purposes of validation.

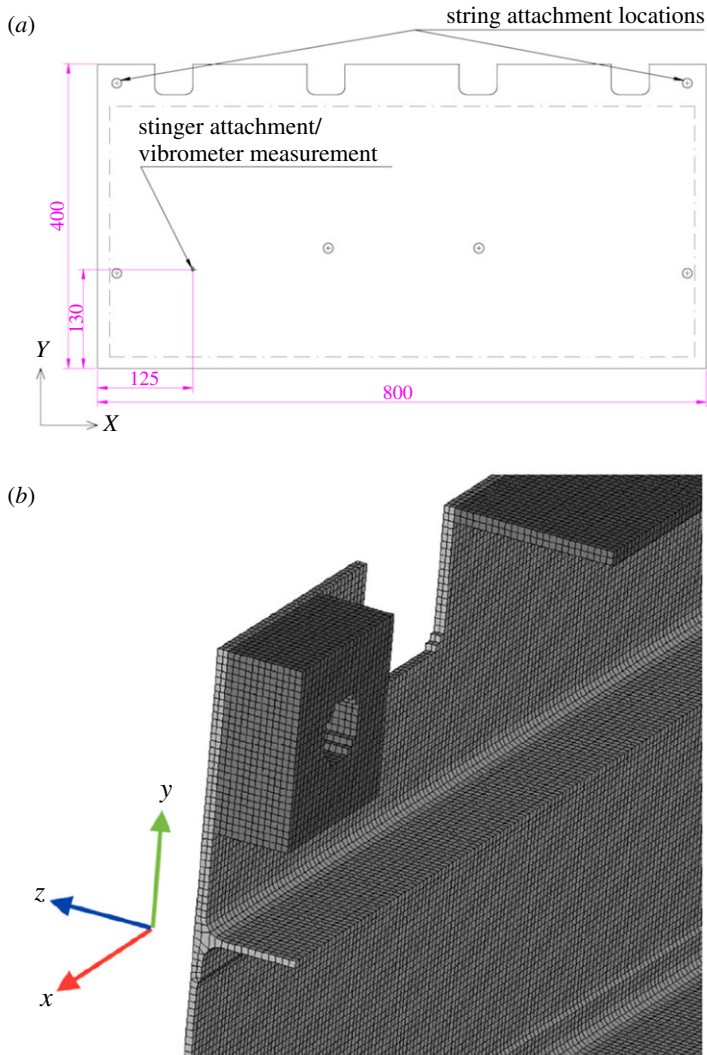

**Figure 5.** Drawing of the front of the panel showing key dimensions and attachment locations (*a*) and a view of part of the simulation mesh shown from behind (*b*), taken from [21].

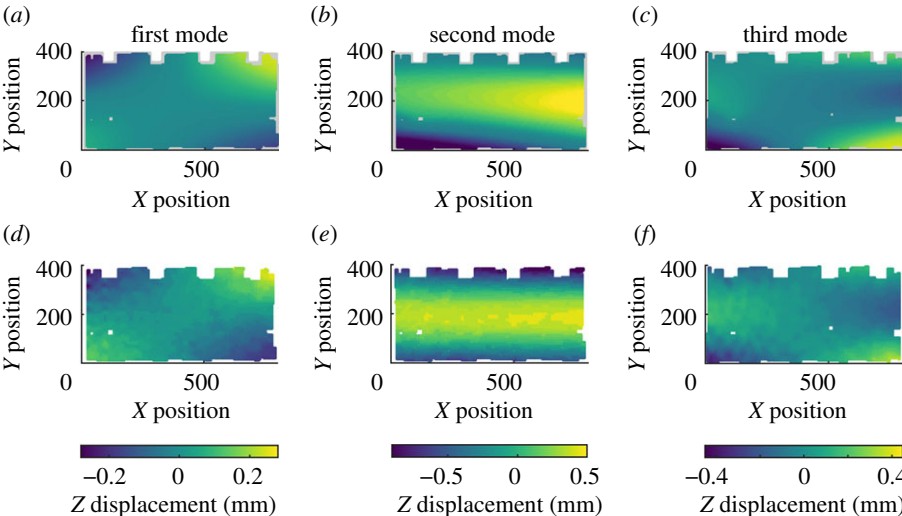

**Figure 6.** Simulated mode shapes (*a*–*c*) and experimental mode shapes (*d*–*f*) for the aerospace panel. Each column depicts data for a different vibrational mode.

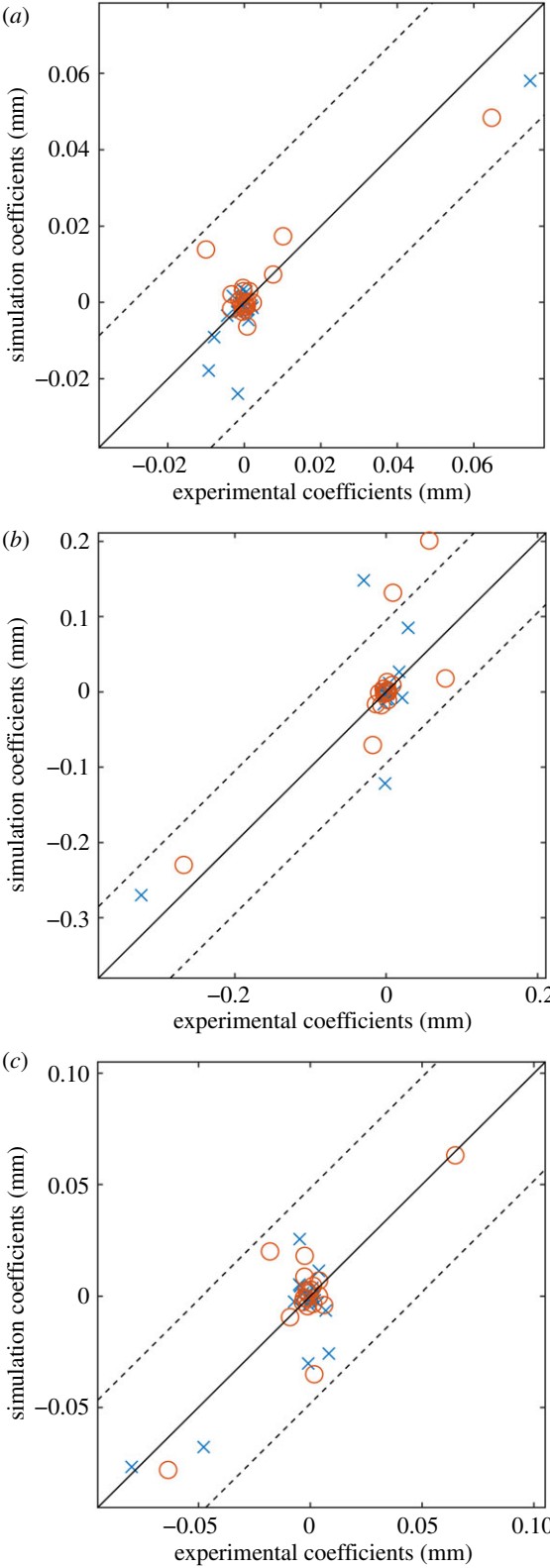

**Figure 7.** Validation diagrams showing the agreement between the mode shapes from the experiment and its simulation for the first (*a*), second (*b*) and third (*c*) modes. The dashed lines indicate the acceptance region that should contain all of the points if the simulation was to be considered acceptable. The circles represent coefficients calculated using Chebyshev decomposition, while the crosses represent coefficients calculated using the new algorithm.

After decomposition, each pair of feature vectors for the three modes were used to perform validation processes. The validation diagrams for the first and third mode shapes, at the top and bottom of figure 7, show that all of the points, represented by crosses, lie within the acceptance region and thus the model

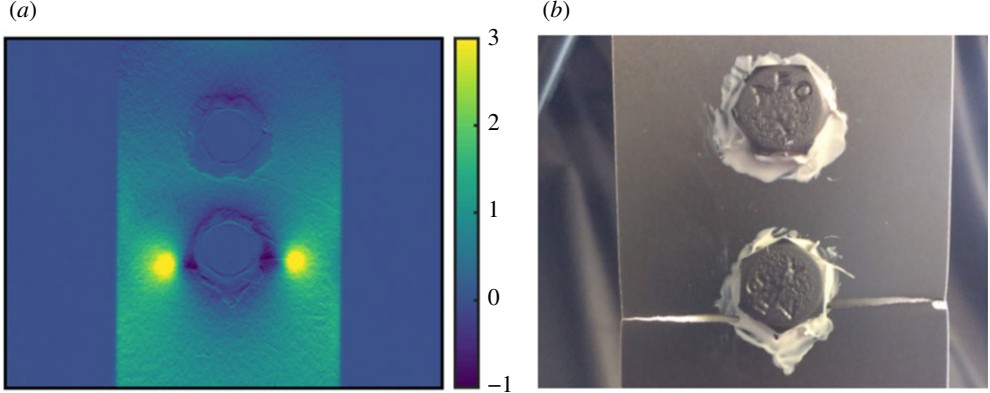

**Figure 8.** A normalized stress field on the surface of the bolted specimen shortly before failure (*a*) and a photograph of the same specimen just after failure (*b*), modified version of a figure in [23].

can be considered acceptable for these modes. However, two points on the diagram for the second mode are outside of the acceptance region and thus based on the CEN methodology, the model cannot be considered acceptable to simulate the second mode of vibration. To compare the new decomposition algorithm with previous methods, the rectangular region shown in the top of figure 5 was decomposed using discrete Chebyshev polynomials, with the results shown by the circles in figure 6. For the first and third mode shapes, all of the Chebyshev coefficients were within the acceptance region. Two of the Chebyshev coefficients for the second mode shape were outside of the acceptance region, which concurs with results obtained using the new algorithm.

## 3.3. Fatigue of a bolted aluminium specimen

In the previous two case-studies, comparisons between datasets were made as part of validation processes. Image decomposition can also be applied for the purpose of condition monitoring and damage assessment [22]. To explore the potential of the new decomposition algorithm for condition monitoring, data from a fatigue study on a bolted aluminium specimen has been used. The specimen was a double lap-joint bolted together at two locations [23,24], shown in figure 8. Sealant was roughly applied around the bolt heads to simulate the appearance of fasteners on aircraft structures. The specimen was then sinusoidally loaded in tension-tension loading until fatigue cracks grew from one of the bolt holes and eventually fractured the specimen.

A thermoelastic stress analysis (TSA) system was used to measure the first stress invariant on the surface of the specimen during fatigue loading. The system was set up to record the stress field every thousand cycles, with the acquisition frequency increased to one capture every 75 cycles after the operator observed changes in the stress field around the bolts. This stress field was then normalized by dividing it by its mean value. In the previous study, the data were not decomposed, instead an optical flow algorithm was used to determine the location of the crack tip as the fatigue crack grew [23]. In recent work, a technique has been developed that uses Chebyshev decomposition to quantify changes to the stress on the surface of a specimen as it is fatigued [25]. It would not be possible to directly apply this technique to the data shown in figure 8 as the stress field around the bolts was obscured by the sealant and thus needed to be removed. This resulted in a rectangular area with an irregular polygonal hole at its centre, shown in figure 9, that cannot be decomposed using Chebyshev polynomials. As the new decomposition algorithm was not limited to rectangular datasets, it was used to decompose all of the data from the bolted specimen, except for the bolt-heads and areas covered by sealant. When performing condition monitoring, it is not necessary to have an exact representation of the original dataset as long as the representation is of a consistent quality [25]. Therefore, monomials up to an order of 10 were used as this order was previously found to be adequate for condition monitoring when working with Chebyshev polynomials [25]. Each stress field was decomposed using the new decomposition algorithm, with the first stress field measured at the start of the experiment used as the first dataset in the algorithm described in §2.2, thus serving as the reference state against which each subsequent stress field was compared. This resulted in pairs of feature vectors, representing the reference stress field and the stress field after a particular number of cycles. The difference between the feature vectors was then calculated using the Euclidean distance

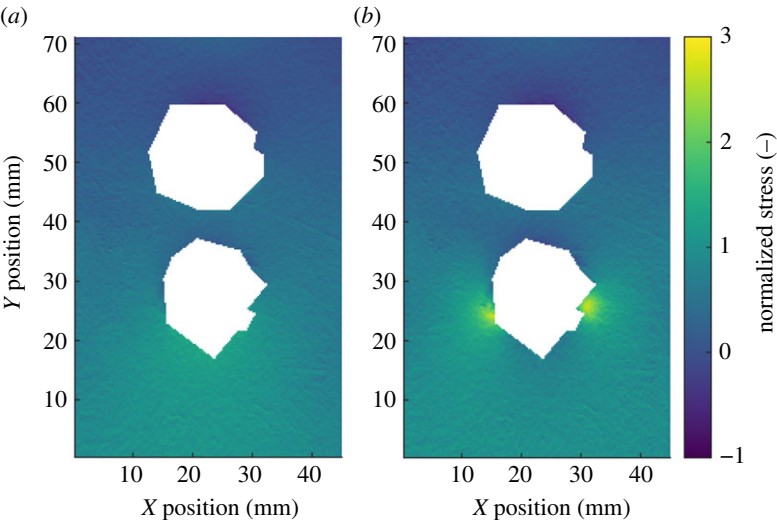

**Figure 9.** The cropped normalized stress fields at the start of the test (*a*) and just after the operator identified the initiation of fatigue cracks (*b*).

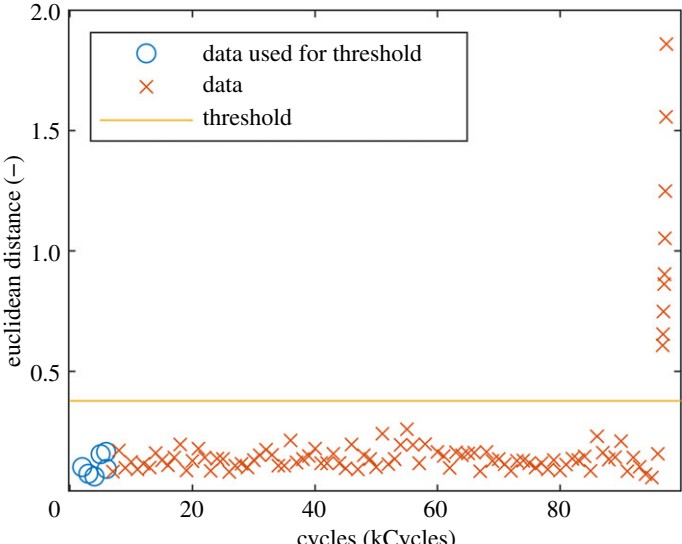

**Figure 10.** Euclidean distances between the feature vectors representing the reference stress field and the stress fields captured after the application of fatigue cycles. The horizontal line indicates the threshold above which a distance is significantly different from the first six values.

between them to obtain a metric describing how much the stress field had changed after the application of fatigue cycles, as shown in figure 10.

The Euclidean distance is nominally constant until crack initiation and rapidly increases thereafter. The variation in Euclidean distance before initiation is principally due to the measurement uncertainty of the TSA system and thus a threshold can be calculated above which changes can be considered statistically significant. The threshold value was computed using simple prediction intervals as:

$$\text{threshold} = \bar{d} + t \times s_d \sqrt{1 + \frac{1}{n}}, \tag{3.1}$$

where $\bar{d}$ is the mean of the first $n$ Euclidean distances, $s_d$ is the sample standard deviation of the first $n$ distances and $t$ is the 99.9th percentile of the t-distribution with $n-1$ degrees of freedom. Measurement noise will only be expected to cause 1 in 1000 Euclidean distances to be greater than this threshold without some underlying change occurring to the stress field. For the purpose of calculating the threshold in figure 10 only the first six Euclidean distances were used. The first Euclidean distance

greater than this threshold corresponded to the normalized stress field captured immediately after the operator observed a change to the uncalibrated stress field in the live view of the TSA system [23].

# 4. Discussion

The three case studies demonstrate the capabilities of the new decomposition algorithm in a range of applications. Different data sources have been used for each of the case studies: a conventional high-speed digital image correlation (DIC) system for the bonnet liner, a pulsed-laser DIC system for the mode shape data and a TSA system for the bolted joints. This shows that the decomposition algorithm can be used to process data from different types of experiments and sources, underlining the wide range of applications for decomposition-based techniques. The concept of basis transformations are well known in numerical computing, with least-squares fitting being the most common application [13]. This study is the first to report the use of these methods for comparing engineering datasets that have irregular shapes, such as those commonly encountered in industry.

Other decomposition techniques have previously been used to process irregular shaped datasets. Singular value decomposition is one such technique that has been used for reducing the dimensionality of deformation data on irregularly shaped components [5]; however, the method still requires the datapoints to be on the same domain for the feature vectors to be comparable. The most similar technique to the proposed approach is AGMDs [7]. The new decomposition technique differs from AGMDs in that it does not require the analyst to tailor the decomposition to the particular shape of the datasets. This makes it easier to automate the comparison process resulting in a potentially more appealing technique for comparing datasets, particularly for those obtained in industry. A further benefit is that the new algorithm does not require interpolation of data, and thus reduces the amount of processing that is done to both datasets during comparisons. The new decomposition algorithm is based on QR factorization which uses Householder reflections to obtain an orthogonal basis with which to decompose the data. This is more numerically stable than Gram-Schmidt orthogonalization used by the AGMD approach. This makes the decomposition more stable when decomposing complex data that requires a large number of coefficients. The QR factorization-based decomposition method also requires less pre-processing of the data prior to decomposition and thus allows for greater automation of the process. In parallel to this study, a software package has been developed so that all of the algorithms described herein can be applied to data with minimal expertise. This package is freely available online and can be used for: aligning, identifying overlapping regions, decomposing, and finally comparing datasets using the CEN validation methodology [26].

Other approaches to irregular shaped data have included interpolating over the area where data is not present so that decomposition can be performed using predefined orthogonal polynomials such as Chebyshev or Zernike polynomials [9]. This means that the comparisons are not simply of experimental and simulation data but also include pseudo-data generated, for example by interpolation. Interpolated areas of data-fields tend to appear smooth with few defining shape features; thus, when quantified comparisons are made, the similarity between the two datasets could increase. This could have the effect of a simulation being declared acceptable when the experimental data actually suggests the simulation is underperforming. It could also mean that condition monitoring misses changes in stress or strain due to damage initiation. This is because changes to the stress or strain field due to damage initiation tend to be subtle, so would be attenuated due to interpolated data that stays nominally constant over time.

A method for decomposing irregular shaped data that does not involve interpolation is to only decompose areas that are of the same shape as the domain of the orthogonal polynomials used for the decomposition. This was the approach used in the earlier study that generated the modal analysis datasets [21]. The rectangular region shown by the dashed line at the top of figure 5 was decomposed using Chebyshev polynomials while the rest of the data was discarded. This simplified the comparison process, but is not desirable for two reasons. First, while a simulation may indicate that the regions to be discarded are free of stress concentrations, it cannot be known *a priori* whether these predictions are correct without them first being validated. By using more of both datasets in the second case study, all of the experimental data was used and thus the model was more rigorously tested. Second, experimental work requires substantial investments of both money and time. Thus, ignoring a subset of the experimental data is essentially an ineffective use of resources. Therefore, while the validation diagrams shown in figure 7 show the same level of agreement for the two approaches, greater confidence can be gained in the validation procedure when using the coefficients calculated with the new decomposition algorithm.

When making comparisons, it is important to ensure that the data used is present in both datasets. Locations where data is available in one dataset but not in another are common. For example, computer simulations can generate predictions of stress or strain at every location on the surface of a component, however optical measurement techniques such as TSA or DIC can only capture data on surfaces that are visible to the camera systems. Thus, if other components or instrumentation block the view of the component being studied, then large areas of simulation data will have no corresponding experimental data. This situation can also occur during condition monitoring, as surface damage may cause coatings on components to flake off, resulting in the measurement system being unable to measure stress or strain at particular locations. This is the cause of the patch of missing data at the centre of the bonnet liner at the bottom of figure 2, at approximately $x = 900$ mm and $y = 400$ mm. It is important that areas of data that exist in one dataset but not the other do not affect the comparison, as without prior knowledge the operator will have no evidence that this additional data is correct. The pre-processing algorithm described in §2.1 quickly identifies and removes this data without the experimenter having to make any subjective judgements. For the bonnet liner case study, this results in parts of the simulation data being removed, shown in grey at the top of figure 2, while no experimental datapoints are lost. When performing validation, the operator must consider whether the quantity of data remaining after alignment is sufficient for making a decision about the validity of the simulation. If only a small proportion of the simulation dataset is used during validation then confidence in decisions will be lower and thus more experimental data should be captured.

The new decomposition technique has been shown to integrate well with the CEN workshop agreement on validation, with no modification to the validation procedure required once the feature vectors and representational error have been obtained. In the modal analysis case study, the validation procedure indicated that the simulation was able to predict the first and third mode shapes but the simulated second mode shape was not acceptable. The second mode shape had the largest displacements and thus would likely magnify any errors in the simulation. This discrepancy was also encountered in the original study, which focused on data from the rectangular portion of the panel, and was attributed to differences in the damping of the panel [21] that had been simulated in a simplistic way. The coefficients calculated in the original study were calculated using discrete Chebyshev decomposition of a rectangular region on the surface of the panel, these coefficients were plotted on the same axes as the coefficients from the new algorithm, shown in figure 7. From this figure, it can be observed that both sets of coefficients had similar magnitudes to those calculated using the new algorithm. This is because both algorithms scale their feature vectors in the same way. A similar observation can be made for the bonnet liner case study, where the AGMD coefficients from the prior study were plotted on the same axis in figure 4. This underlines that while the approach to calculating the coefficients is different to preceding techniques, it can replace these techniques without any further changes needing to be made to the validation procedure.

It is common for experiments to be manually monitored to observe when significant changes occur. For example, while conducting full-scale structural tests, manufacturers monitor for damage initiation so that its possible locations can be determined before it occurs in structures in service. For the testing of a large structure, this can require significant amounts of skilled labour. For the third case study, a bolted specimen loaded in tension-tension fatigue was considered. The measured stress field on the surface of the specimen was manually monitored by an operator watching for changes to the stress field on the portions of the specimen that were not covered in sealant. During the test, the operator only focused on areas where good data was available. Similarly, the decomposition algorithm only used these areas, with locations containing poor data masked. While this damage monitoring approach has been explored in previous studies [25], the new damage assessment algorithm increases its readiness for application in industry, where components with complicated shapes are common.

In this study, the new decomposition algorithm has been applied to displacement and stress data. However, it is not limited to only processing this kind of data. The only requirements are that the datapoints can be projected onto a plane and that the measurand at each datapoint is represented by a scalar real number. Therefore, the algorithm could be applied to data from techniques outside of the field of applied mechanics. With some modifications it could also be used to decompose irregularly shaped three-dimensional datasets such as from computed tomography. When combined with the workshop agreement on validation, this new decomposition algorithm provides a consistent approach to comparing data-fields, regardless of their origin and the purpose of the comparison. This consistency removes subjective judgements and thus provides greater confidence when comparisons are made for quantifying the quality of computer simulations or for condition monitoring.

# 5. Conclusion

An algorithm for decomposing data has been developed that can be used for the comparison of datasets with irregular shapes in applied mechanics. This enables decomposition-based validation procedures to be applied even if there are significant differences between the positions of datapoints in the two datasets, or if there are large holes due to missing data or complicated component geometries. The algorithm also improves condition monitoring for parts with complex geometries by focusing the assessment on locations where high-quality data is available. The decomposition algorithm was applied to data from a range of different experiment types and data sources. Data from finite-element analysis, digital image correlation and TSA were processed without the need for any modification to the algorithm, demonstrating that it is a robust and widely applicable approach. The decomposition algorithm eliminates the need for any interpolation of data during comparisons and requires few decisions by the operator during data processing. This could lead to more straightforward validation procedures and automation of condition monitoring that require less expertise from the operator.

Data accessibility. The simulation and experimental data are available through the Dryad Data Repository [27].

Authors' contributions. A.D.D. and W.J.R.C. developed the algorithm. W.J.R.C. and A.D.D. wrote the first draft of the manuscript. W.J.R.C., K.D. and E.A.P. supervised the project. C.A.M. captured one of the datasets and advised on its processing. All authors contributed to the final manuscript.

Competing interests. We have no competing interests.

Funding. This research has received funding from the Clean Sky 2 Joint Undertaking under the European Union's Horizon 2020 research and innovation programme under grant agreement no. 754660:MOTIVATE (Matrix Optimisation for Testing by Interaction of Virtual And Testing Environments).

Acknowledgements. The authors are grateful for the work of Brodie Payter and Jacob Curtis, both undergraduate students at the University of Liverpool, on the development of the software tool, Theon, which implements the algorithms described in §2 and was used to process the data in this manuscript. Figure 1 is reused with permission from [20]. Figure 5 was made by combining two figures reused with permission from [21]. Figure 8 is a modified version of a figure in [23] and is used under the terms of a Creative Commons Attribution 4.0 International License (http://creativecommons.org/licenses/by/4.0/).

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
