## [Peer Review File · Royal Society Open Science]

Review History

RSOS-210916.R0 (Original submission)

Review form: Reviewer 1

Is the manuscript scientifically sound in its present form?

No

Are the interpretations and conclusions justified by the results?

Yes

Is the language acceptable?

Yes

Do you have any ethical concerns with this paper?

No

Have you any concerns about statistical analyses in this paper?

No

Recommendation?

Major revision is needed (please make suggestions in comments)

Comments to the Author(s)

In this paper, a decomposition algorithm based on QR factorization is proposed for processing and comparing irregularly shaped stress and deformation data obtained from structural analysis. The results show that the proposed method is feasible. Meanwhile, the authors should consider these following points for this work.

1. The novelty of the proposed method should be considered and put in detail since Gram-Schmidt method is a common way to get orthogonal basis and is widely used to achieve QR factorization.
2. There is an obvious contradiction in that the author said P is the matrix of K+1 basis vectors with a dimension of $N \times k$.
3. The author mentioned singular value decomposition as a method to compare data. The work would be more thoroughly explored if the results of SVD and QR decomposition are compared since SVD is a universally used method to reduce dimensionality.
4. "Gram-Schmidt" was misspelled. The expression "Another approach to missing data..." in section 4 is confused.

Review form: Reviewer 2

Is the manuscript scientifically sound in its present form?

Yes

Are the interpretations and conclusions justified by the results?

No

Is the language acceptable?

Yes

Do you have any ethical concerns with this paper?

No

Have you any concerns about statistical analyses in this paper?

No

Recommendation?

Accept with minor revision (please list in comments)

Comments to the Author(s)

This paper investigates algorithms for comparing full-field 3D data sets from objects with complex shape.

1. If I understand correctly, the authors propose that their new decomposition algorithm which is introduced in this paper has a better performance than existing methods, such as Zernike or Chebyshev polynomials due to the limitations of the 2D shape of the measurement data.
2. Pages 2, line 45 to page 3, line 13. This paragraph is missing references to many of the statements made when introducing new topics, including in the first, second, fourth and fifth sentences.
3. Pages 3-6, section 2. It would be helpful to give an overview of the numerical method at the start of section 2 to guide the reader through the detailed technical content.

4. Pages 3-6, section 2. The numerical method section is quite descriptive and it is not clear which parts of the methodology have been developed by the authors and which are standard procedures. Also the choices made in determining the methodology should be justified better, for example:

a. Page 3, lines 47-48. Why was manual translation and rotation deemed sufficient?

b. Page 4, lines 3-6. Why in particular was the MATLAB alpha shape algorithm chosen and what is its performance and relative resolution, relative to the resolution of the dataset, of the MATLAB alpha shape algorithm for identifying the outer boundary, or hole, in a shape?

5. Referring to the case studies in section 3, the second example, the aerospace panel modal analysis does show a comparison with Chebyshev decomposition in figure 7, but both algorithms have the same number of outliers (two) and only for the second mode. How does this demonstrate the better performance of the proposed algorithm?

Overall this is a good quality paper, but it needs more justification of methods and a more careful evaluation of results.

Decision letter (RSOS-210916.R0)

Dear Dr Christian

The Editors assigned to your paper RSOS-210916 "Comparing full-field data from structural components with complicated geometries" have now received comments from reviewers and would like you to revise the paper in accordance with the reviewer comments and any comments from the Editors. Please note this decision does not guarantee eventual acceptance.

Please submit your revised manuscript and required files (see below) no later than 21 days from today's (ie 08-Jul-2021) date. Note: the ScholarOne system will 'lock' if submission of the revision is attempted 21 or more days after the deadline. If you do not think you will be able to meet this deadline please contact the editorial office immediately.

on behalf of Dr Adil Al-Mayah (Associate Editor) and R. Kerry Rowe (Subject Editor)
 openscience@royalsociety.org

Associate Editor Comments to Author (Dr Adil Al-Mayah):

Associate Editor: 1

Comments to the Author:

A number of critical points have been raised by the reviewers regarding the method details, and results evaluations. Although, the feasibility of the presented technique is demonstrated using three case studies, adding more details of the technique and validation of the results is recommended.

Reviewer comments to Author:

Reviewer: 1

Comments to the Author(s)

In this paper, a decomposition algorithm based on QR factorization is proposed for processing and comparing irregularly shaped stress and deformation data obtained from structural analysis. The results show that the proposed method is feasible. Meanwhile, the authors should consider these following points for this work.

1. The novelty of the proposed method should be considered and put in detail since Gram-Schmidt method is a common way to get orthogonal basis and is widely used to achieve QR factorization.
2. There is an obvious contradiction in that the author said P is the matrix of $K+1$ basis vectors with a dimension of $N \times k$.
3. The author mentioned singular value decomposition as a method to compare data. The work would be more thoroughly explored if the results of SVD and QR decomposition are compared since SVD is a universally used method to reduce dimensionality.
4. "Gram-Schmidt" was misspelled. The expression "Another approach to missing data..." in section 4 is confused.

Reviewer: 2

Comments to the Author(s)

This paper investigates algorithms for comparing full-field 3D data sets from objects with complex shape.

1. If I understand correctly, the authors propose that their new decomposition algorithm which is introduced in this paper has a better performance than existing methods, such as Zernike or Chebyshev polynomials due to the limitations of the 2D shape of the measurement data.
2. Pages 2, line 45 to page 3, line 13. This paragraph is missing references to many of the statements made when introducing new topics, including in the first, second, fourth and fifth sentences.
3. Pages 3-6, section 2. It would be helpful to give an overview of the numerical method at the start of section 2 to guide the reader through the detailed technical content.

4. Pages 3-6, section 2. The numerical method section is quite descriptive and it is not clear which parts of the methodology have been developed by the authors and which are standard procedures. Also the choices made in determining the methodology should be justified better, for example:

- a. Page 3, lines 47-48. Why was manual translation and rotation deemed sufficient?
- b. Page 4, lines 3-6. Why in particular was the MATLAB alpha shape algorithm chosen and what is its performance and relative resolution, relative to the resolution of the dataset, of the MATLAB alpha shape algorithm for identifying the outer boundary, or hole, in a shape?

5. Referring to the case studies in section 3, the second example, the aerospace panel modal analysis does show a comparison with Chebyshev decomposition in figure 7, but both algorithms have the same number of outliers (two) and only for the second mode. How does this demonstrate the better performance of the proposed algorithm?

Overall this is a good quality paper, but it needs more justification of methods and a more careful evaluation of results.

===PREPARING YOUR MANUSCRIPT===

===PREPARING YOUR REVISION IN SCHOLARONE===

Author's Response to Decision Letter for (RSOS-210916.R0)

See Appendix A.

Decision letter (RSOS-210916.R1)

Dear Dr Christian,

It is a pleasure to accept your manuscript entitled "Comparing full-field data from structural components with complicated geometries" in its current form for publication in Royal Society Open Science. The comments of the reviewer(s) who reviewed your manuscript are included at the foot of this letter.

on behalf of Dr Adil Al-Mayah (Associate Editor) and R. Kerry Rowe (Subject Editor)
openscience@royalsociety.org

Appendix A

Dear Adil Al-Mayah,

Thank you for your comments and the comments of the reviewers. We have responded to each comment explaining how we have edited the manuscript to reflect their suggestions.

Comments from associate Editor

A number of critical points have been raised by the reviewers regarding the method details, and results evaluations. Although, the feasibility of the presented technique is demonstrated using three case studies, adding more details of the technique and validation of the results is recommended.

- Thank you for this suggestion we have added additional details during the process of responding to all of the reviewers' comments.

Reviewer 1

1. The novelty of the proposed method should be considered and put in detail since Gram-Schmidt method is a common way to get orthogonal basis and is widely used to achieve QR factorization.

- We accept that QR factorisation is a common technique in mathematics and that the novelty of the methods in this manuscript is how QR factorisation is applied to comparison of engineering data. A statement explaining this has been added to Section 4.

2. There is an obvious contradiction in that the author said P is the matrix of $K+1$ basis vectors with a dimension of $N \times k$.

- This has been corrected, as this mistake was only in the manuscript it has no effect on any of the results.

3. The author mentioned singular value decomposition as a method to compare data. The work would be more thoroughly explored if the results of SVD and QR decomposition are compared since SVD is a universally used method to reduce dimensionality.

- It would be interesting to compare these two techniques in detail however in order to use SVD the datasets would need to be on the same domain for the feature vectors to be comparable which is not possible for the bonnet liner and aerospace panel case studies. A statement considering this has been added to Section 4.

4. "Gram-Schmidt" was misspelled. The expression "Another approach to missing data..." in section 4 is confused.

- The spelling mistake has been corrected at all four locations. The sentence in Section 4 has been rewritten.

Reviewer 2

1. If I understand correctly, the authors propose that their new decomposition algorithm which is introduced in this paper has a better performance than existing methods, such as Zernike or Chebyshev polynomials due to the limitations of the 2D shape of the measurement data.

- This is correct, we have modified the summary to make this clearer.

2. Pages 2, line 45 to page 3, line 13. This paragraph is missing references to many of the statements made when introducing new topics, including in the first, second, fourth and fifth sentences.

- We have added new references and re-cited some papers in this paragraph and the next to ensure the topics are suitably referenced.

3. Pages 3-6, section 2. It would be helpful to give an overview of the numerical method at the start of section 2 to guide the reader through the detailed technical content.

- An overview of the numerical method is now provided in section 2.

4. Pages 3-6, section 2. The numerical method section is quite descriptive and it is not clear which parts of the methodology have been developed by the authors and which are standard procedures.

The concept of QR factorisation is well developed in mathematics

- A statement has been added to Section 4 making it clear that QR factorisation is commonly used in mathematics and that it is its application to the comparison of solid-mechanics data that is novel.

Also the choices made in determining the methodology should be justified better, for example:

a. Page 3, lines 47-48. Why was manual translation and rotation deemed sufficient?

- Statements have been added in section 2 to explain more clearly why this method was used.

b. Page 4, lines 3-6. Why in particular was the MATLAB alpha shape algorithm chosen and what is its performance and relative resolution, relative to the resolution of the dataset, of the MATLAB alpha shape algorithm for identifying the outer boundary, or hole, in a shape?

- Further information about the choice of boundary identification algorithm has been added to Section 2.

5. Referring to the case studies in section 3, the second example, the aerospace panel modal analysis does show a comparison with Chebyshev decomposition in figure 7, but both algorithms have the same number of outliers (two) and only for the second mode. How does this demonstrate the better performance of the proposed algorithm?

- A statement has been added to section 4 drawing the reader's attention to the similarity and explaining how the new algorithm increases the user's confidence in the validation procedure.

Overall this is a good quality paper, but it needs more justification of methods and a more careful evaluation of results.

- We are grateful for the comments and are confident that the additions made for both reviewers satisfy the need for further justification and evaluation.